# The Role of Working Memory in Early Literacy and Numeracy Skills in Kindergarten and First Grade

**DOI:** 10.3390/children10081285

**Published:** 2023-07-26

**Authors:** Marina Shvartsman, Shelley Shaul

**Affiliations:** Edmond J. Safra Brain Research Center for the Study of Learning Disabilities, Department of Learning Disabilities, Faculty of Education, University of Haifa, Haifa 3103301, Israel; marvadsh@gmail.com

**Keywords:** early literacy, early numeracy, working memory, kindergarten, first grade

## Abstract

The working memory system supports learning processes such as acquiring new information and the development of new skills. Working memory has been found to be related to both early literacy and early numeracy in kindergarten and to linguistic and mathematical academic skills at older ages, but the contribution of each of the memory components at these ages is not yet clear. The purpose of this study is to examine the unique connections among the various systems of WM, early literacy, and early numeracy using various assessment tests of simple WM and complex WM, as well as a variety of tasks in math and language skills administered to the same 250 children in kindergarten and 150 children in first grade. Consistent with the predictions, significant relations among all components of memory and mathematics and language knowledge at both ages were found, although these connections were differential for the different types of tasks and memory systems. The connection of complex WM was stronger in its contribution and more significant in first grade in both mathematics and language domains. Complex WM resources were more important in early literacy at kindergarten age, while simple WM seems to be important in early numeracy. The theoretical and educational implications of these results are discussed accordingly.

## 1. The Role of Working Memory in Early Literacy and Numeracy Skills in Kindergarten and First Grade

### 1.1. Rationale

Working memory has a central role in supporting learning processes, such as mastering new information and the development of new skills [1,2]. Learning processes require simultaneous processing and storage of data; therefore, children with memory disabilities display difficulties in basic learning processes, in acquiring new knowledge, and in performing complex skills [3].

Recently, more researchers have attempted to explain children’s development and performance in the fields of language and mathematics by focusing on both domain-general cognitive functions and domain-specific components (e.g., [4,5,6]). It is agreed that WM is substantially correlated with the development of language skills and mathematical skills, but the individual contribution of each of the memory components is not yet clear, due to the limited studies among kindergarten and first grade children, and the lack of consistency in findings in the various fields of WM.

The aim of the current study is to examine the unique correlations between the various systems of working memory (WM), early literacy, and early numeracy in kindergarten, in addition to the examination of the developmental process of these abilities in first grade.

### 1.2. Theoretical Background

#### 1.2.1. Working Memory

The WM system has a significant role in supporting learning. Most of the studies which have examined the cognitive processes that underlie arithmetical and language abilities has used the multicomponent model of WM which was presented by Baddeley and Hitch in 1974 and updated in later years [1,7]. According to this model, WM is a multicomponent system. There are specific short-term memory (STM) stores, the phonological loop and the visuospatial sketchpad; their role is to maintain verbal information and visual and spatial information, respectively. The central executive (CE) system controls and navigates the cognitive processes, especially when performing two or more tasks simultaneously, and enables flexible use of the two short-term memory systems according to arising needs. Different WM capacities influence the growth of knowledge and new skills [8] and are associated with academic and cognitive performance [2,9].

#### 1.2.2. Development of Mathematical Skills

Several theoretical models have been suggested in order to explain the developmental process of acquiring mathematical skills. Dehaene proposed a Triple Code Model [10], which claims that numerical processing involves three types of mental representation: two symbolic forms in which numbers are represented as words (e.g., “three”) and Arabic numerals (e.g., “3”) and a non-symbolic system which represents numbers as magnitudes (quantities). It has been proposed that children are born with basic cognitive skills which process non-symbolic quantity information (“analogue code”). When children start formal schooling, they learn the number words and symbols which are mapped onto their corresponding quantity, assuming that the non-symbolic skills provide meaning to number symbols which are later used in different math tasks [10].

The Core Numerical Skills Model offers another explanation for the foundation of early numeracy skills. This model, which was based on longitudinal studies, suggests that there are individual differences in early mathematical skill development in five- to eight-year-old children. The Core Numerical Skills Model divides the skills into four categories: symbolic/non-symbolic number sense, understanding mathematical relations, counting skills, and basic skills in arithmetic. The symbolic and non-symbolic skills are defined as processes which are used to approximate the magnitudes of symbols. Mathematical relations skills in the Core Numerical Skills Model overlap with early numeracy skills; one example is understanding numerical relations. Furthermore, mathematical relations skills include mathematical-logical principles such as understanding the meaning of the base-10 system [11]. The next component of this model is counting skills. ‘Counting skills’ refers to the child’s knowledge of number symbols and their sequence, such as counting forwards and backwards; and enumeration [11]. Basic arithmetic skills constitute the fourth component, and in five- to eight-year-olds include the mastery of addition and subtraction tasks [11]. The model was empirically tested [12]. These factors are different skills that may be highly related. This model has been chosen as the theoretical framework for the current study.

#### 1.2.3. Development of Language Skills

The term “early literacy” refers to the first steps young children take in understanding the written world, prior to formal education in reading and writing. Early literacy includes several components in three major areas: spoken language, phonological awareness, and recognition of printed words [13,14]. Phonological awareness fully develops after the child learns how to read and write, not in a spontaneous manner; as children become more aware of smaller parts of words, they can manipulate them [15,16].The acquisition of alphabetic knowledge and letter–sound relationships develop together; these abilities are related to each other and are of vital importance for learning how to read and write. [17] Stated that alphabetic knowledge as well as the recognition of letters and their sounds are vital for the acquisition of reading and writing. Skills related to print knowledge were found to be predictors of literacy skills such as reading and spelling. Exposure to print enhances orthographic learning of specific words as well as morpho-orthographic knowledge. In the first stages of learning how to read, children mainly use bottom-up sequential spelling-sound decoding, whereas the older, more skilled readers rely mainly on lexical-orthographic features of the word. Therefore, phonological decoding is used less after first grade; at this later stage, reading mainly relies on the word level, which is based on morphological and lexical information [18].

Correlations between children’s mastery of early language skills at kindergarten age and their success in the processes of learning how to read and write in elementary school have been found in different languages [19]. Children who begin school without basic skills may lag behind their peers. It has been found that during the progression of years in formal education, these children exhibit difficulties in bridging the gaps [20]. Early numeracy and early literacy serve as the base for the development of academic abilities at school; therefore, it is vital to understand how these develop and what their cognitive foundation is.

#### 1.2.4. The Contribution of Working Memory to Mathematical Learning

Many researchers have studied the contribution of WM to different mathematical skills among both kindergarteners and primary school children and even older children (e.g., [2,21]. It is widely assumed that WM is a crucial factor in many aspects of learning mathematics [22,23], because it is related to all components of mathematical tasks [24,25]. In the first stages of learning math, connections were found between WM and simple calculations. This was explained by the WM resources which are needed at a younger age, when new mathematical skills and concepts are acquired [26,27]. At older ages, children’s WM abilities were related to remembering instructions, choosing and using different mathematical strategies, and mental calculations, including calculations with multi-digit numbers [28]; dealing with complex aspects of mathematical applications that require taking several steps and coordinating between them, as in solving complex mathematical equations [29]; or performing mathematical tasks with visual components, such as estimation of numbers along the sequence of natural numbers and fractions [24,30]. Greater WM capacity improves the storage quality of word problems and their solutions and strengthens the connection between problems and solutions in long-term memory. In other words, it enables the quick and accurate extraction of mathematical facts [2]. Children with more highly developed WM resources succeed in extracting mathematical facts or connecting between new and old data in a more satisfactory manner compared with children with a less-developed WM [31].

Despite the consensus that WM is essential to studying mathematics, there is no unequivocal understanding of how the components of WM contribute to learning and mastery of various mathematical skills and the way in which the significance of each WM component may vary throughout the learning process. Some studies found that the phonological loop was involved in counting and enumeration [32,33] or counting backwards [21]. (Its effects were also noted on strategies for data acquisition and the extraction of numeric facts from long-term memory [34] and on mental calculation activity [35]. The PL has also been found to be related to automatization and direct extraction strategies for adding and subtracting low numbers and quantities [36] and served as a significant contributory factor in solving word problems among six-year-olds [34], though not as a sole contributory factor. Among older children, the contribution of the PL appears to be significant in solving calculus problems or performing exact mental calculations, when using verbal counting strategies, or when required to transform symbols and numeric models into language codes [21,37].

Less is known regarding the role and contribution of the visuospatial sketchpad. Some researchers claim that the visuospatial sketchpad plays an important role in the mathematical achievements of kindergarten and preschool children [36,38,39], while others maintain that the visuospatial sketchpad has a larger capacity at later ages, therefore playing an important role in manipulating multi-digit numbers as well as complex algebra and geometry problems [25,30]. It has been found that the visuospatial sketchpad plays an important role in representing numbers on the mental number axis [30,32], since spatial-numeric data about the location of numeric symbols is vital to the precise execution of simple counting and enumeration tasks, and young children make use of visuospatial strategies when making mental calculations [40]. In addition, the visuospatial sketchpad has also been found to be related to children’s performance with non-verbal mathematical problems [41].

Similarly, researchers are in dispute regarding the contribution of the central executive to mathematical skills. On the one hand, studies of early numeracy have shown the contribution and influence of the CE on the mathematical performance of kindergarten age children in many skills, which were examined in accordance with the appropriate curriculum for their age group [30]. On the other hand, it has been proposed that comparing numbers, unlike comparing groups of numbers, does not involve the WM, because number groups are already identified and mapped as symbols [42]. They further claim that in less complex skills, WM does not serve as a predictive factor. They argue that WM is involved in early numeracy skills that require a series of steps or data storage (e.g., calculations or number sequences) but is not involved with basic components such as verbal counting and one-to-one correspondence of numbers [42].

It is therefore essential to study the connection between WM and specific components of mathematics throughout the developmental stages of mathematics skills.

#### 1.2.5. The Contribution of Working Memory to Language Learning

Over the years, many studies and theoretical models have dealt with the development of WM and its connection to individual differences in language and reading abilities. Researchers agree that WM plays an important role in the development of language and reading and writing skills [28,43], although there is no consensus regarding the connection between the various memory components and these abilities.

With regard to early literacy skills, various studies found a correlation between phonological WM and phonological awareness and reading abilities [44]. Verbal WM has been defined as a determining factor for reading acquisition both among children with typical development and children with reading disabilities [45,46]. It is claimed that verbal WM influences meta-linguistic abilities including phonological awareness, and it contributes to the long-term learning of the letter-sound rules essential for developing phonological processing abilities, which are a crucial and basic condition for acquiring reading and writing skills [47]. The phonological loop has been found to play an essential role in reading skills [27,48]. Regarding the visual spatial component, some studies show that it does not bear any effect on reading skills [49], while others report this component to be deficient among both children and adults with reading disabilities [45,50]. Some researchers believe that visual spatial memory can compensate for deficient language storage and lower verbal processing ability. As a result, children can use various types of visuospatial representations in order to overcome their difficulties [51].

Whereas many studies deal with the importance of phonological processing procedures as mentioned above, the central executive itself receives less focus in research, and its role is controversial. The connection between the central executive and reading skills is not clear. Some researchers claim that the quality of performance of the central executive processor distinguishes between typical readers and those with disabilities [52]. (However, this component controls and uses information that stems from the phonological loop. As a result, any deficiency in the function of the central executive processor, or a combination of the functions of the phonological loop with that of the central executive processor, may influence language skills [51].

While learning how to read, readers grasp the alphabetic principle and obtain the ability to use knowledge that is based on the relations between spelling and sound in order to correctly decode any new word. The performance of WM at this phase is very significant for phonological coding and for acquiring stable relations between grapheme and phoneme [53]. It has been established that children with low literacy skills (awareness of the alphabetic principle) perform phonological memory tasks at a lower skill level than their peers [54].

In a study conducted by Gathercole and colleagues (2003), the verbal WM abilities of children entering the educational system (at ages four to five) were closely correlated to their achievements at school two-and-a-half years later (at age seven). In other words, children with low WM capacity are at high risk for under-achieving in the first years of school.

In light of the findings described here, memory ability is significant in the early identification of children at risk for different types of learning disabilities. However, due to lack of uniformity in those findings, it is necessary to examine which language and mathematical abilities are related to the performance of specific WM mechanisms, at an early age and in first grade, before difficulties begin to appear.

### 1.3. The Current Study

A primary goal of this study is to examine the contribution of each of the WM mechanisms to early literacy and early numeracy in kindergarten and to math and language achievements in first grade. Another goal is to examine the developmental course of WM to determine whether these connections change once children enter school and start the formal stages of acquisition of the different academic skills. By studying the influence of cognitive processes (which are not specific skills in the field of learning), we can gain a deeper understanding of the cognitive mechanisms underlying various learning tasks and domains. This knowledge can help uncover how information is processed, organized, and utilized in the learning and problem-solving contexts. In addition, investigating the effect of WM on learning processes informs instructional practices, facilitates intervention development, acknowledges individual differences, and supports cognitive development throughout the lifespan. It helps optimize learning experiences and promotes academic success for diverse learners.

Due to fact that simple and complex WM have been found to be connected in children [28,55], we would expect both simple and complex WM measures to relate to performance in the linguistic and mathematical fields. In addition, we expect that performance in the two domains in both age groups will have a stronger connection to complex WM tasks than to simple verbal or visual spatial tasks. Moreover, academic assignments based on verbal components will relate more strongly to verbal WM and academic assignments based on visual spatial components (such as mathematics) would relate more strongly to visual spatial WM, due to the modality-specific subcomponents of the WM [56].

## 2. Method

### 2.1. Participants

Data were collected from the same children at two time points, first in the middle of the kindergarten year and then twelve months later in the middle of first grade. The first stage of the study included 250 kindergarteners between the ages of five and seven years old (*M* = 5.78 years, *SD* = 0.36), 50.8% boys and 49.2% girls. All of the participants were Hebrew-speaking typically developing children with no known developmental disorders. The second stage, a year later, included 137 primary school pupils between the ages of six and eight years old (*M* = 7.19 years, *SD* = 0.45), 48.9% boys and 51.1% girls. Due to the outbreak of COVID-19 and the closure of schools, we were unable to examine some number of the children from the first stage of the study again, during their first grade year. The children in the study were from varied socio-economic backgrounds, attending 30 different kindergartens and 10 schools in the north of Israel.

### 2.2. Measures

#### 2.2.1. Early Numeracy in Kindergarten and Math Skills in First Grade

Sixteen mathematics tasks were used to assess individual numeracy constructs from the different aspects of early mathematical knowledge noted earlier, and nine mathematics tasks were given to first graders. For all tasks, children received one point for each correct response. The tests are partly based on tests created by Purpura and his colleagues [43,57]. All tasks were conducted in a random order which was different for each child.

**Verbal Counting (Forward or Backward).** Kindergarten. The child was asked to count forward from 6 to 20. After one mistake, the task was stopped and the child was asked to count forward from 1 to 20. After one mistake, the task was stopped. Each number word correctly pronounced received one point. The total score was calculated separately for each counting-forward subtest (maximum = 14 or 20). If the child counted correctly from 6 to 20, he or she also received the full score for the series from 1 to 20. The child was asked to count backward from 20 to 0. After one mistake, the task was stopped and the child was asked to count backward from 10 to 0. After one mistake, the task was stopped. Each number word correctly pronounced received one point. The total score was calculated separately for each counting-backward subtest (maximum = 20 or 10). Children who counted correctly from 20 to 0 also received the full score for the series from 10 to 0.

**First Grade**. The child was asked to count forward from 11 to 31. After one mistake the task was stopped. Each number word correctly pronounced received one point. The total score was calculated separately for each counting-forward subtest (maximum = 20). The child was asked to count backward from 20 to 0. After one mistake, the task was stopped and the child was asked to count backward from 32 to 9. After one mistake, the task was stopped. Each number word correctly pronounced received one point.

**Verbal Counting—Skips of 2.** Kindergarten. The examiner started a count sequence (skips of two—forward) and instructed the child to continue to count until 20. After one mistake, the task was stopped. Each number word correctly pronounced received one point. The maximum total score for the counting subtest was 10.

**First Grade**. The examiner started a count sequence (skips of two—forward) and instructed the child to continue to count until 20. After one mistake the task was stopped. Each number word correctly pronounced received one point. The maximum score for the counting subtest was 10. The examiner started a count sequence (skips of two—backward from 20) and instructed the student to continue to count until 0. After one mistake the task was stopped. Each number word correctly pronounced received one point. The maximum score for the counting subtest was 10.

**One-to-One Counting.** Kindergarten. The child was presented with four sets of game discs (3, 6, 11, and 14 discs, respectively) and was asked to count them. Each correct amount was given 1 point. There was a total of four items (α = 0.51).

**Cardinality and Conservation Ability.** Kindergarten. In the context of the one-to-one counting task, the child was asked to put the discs into a transparent glass and indicate how many discs they had counted. The correct answer was the last number he or she counted, without re-counting the set. There were a total of two items (α = 0.54).

**Composition: Disassembly of a Group into Parts.** Both first graders (α = 0.60) and kindergarteners (α = 0.54) were examined using these tests. The child was presented with two sets of game discs (6 and 9 discs, respectively) and was asked to separate the quantity into two groups in two different ways. For each set that he or she arranged correctly, the child scored one point.

**Subitizing**. Kindergarten. The child was briefly presented with a set of dots (2, 4, 3, 5) in a linear fashion and was instructed to say how many items were in each group (α = 0.43).

**Ordinality.** Kindergarten. The child was presented with a line of game discs and was asked to identify “last”, “first”, “before last” and the 10th disc. There were a total of six items (α = 0.51).

**Symbolic and Non-Symbolic Magnitude Comparison** based on the numeracy screener test. Both first graders and kindergarteners were examined by these tests. In the symbolic magnitude comparison, the children were required to identify the biggest number in each single-digit numerical pair. In the non-symbolic magnitude comparison, the children were required to recognize the larger magnitude of two arrays of dots without counting. In both subtests, the children marked their choice with a pencil as quickly and accurately as possible. A total of correct answers given within a one-minute time limit was calculated for each subtest. Test–retest reliability for symbolic comparison was 0.96, and for non-symbolic comparison, 0.94.

**Numeral Identification.** Kindergarten. The child was presented with a card with five numbers (3, 6, 4, 9, 5). The examiner said a number and the child had to point to the number he or she heard. The numbers were said in a random order (α = 0.77).

**Number Naming.** Kindergarten. The child was required to verbally name 13 numbers (from 0 to 12). The numbers were presented in random order (α = 0.89). First Grade. Children were required to verbally name 15 numbers (between 0 and 40). The numbers were presented in random order (α = 0.89).

**Number to Quantity Matching.** Kindergarten. The child was asked to match a picture showing a certain number of objects with the correct number from 1 to 4 printed on a card. There was a total of five items (α = 0.56).

**Number Order.** Kindergarten. The child was asked to arrange cards numbered 0 to 10, from the smallest number up. Each number placed in the correct sequence entitled participants to one point. The overall score was the total of all points scored.

**Formal Addition.** Kindergarten. The task included simple single-digit addition sums of numbers between 1 and 5 with a maximum sum of 5 (e.g., 1 + 1, 3 + 2). The child was presented with the sum orally and was asked, ‘‘How much is…?” There were a total of five items (α = 0.79).

**First Grade**. The task included 30 simple sums with numbers between 1 and 10; the children were asked to answer as many as sums as possible within a time limit of 120 s (α = 0.94).

**Formal Subtraction.** Kindergarten. The task included simple single-digit subtraction sums, when the sum of parts of the whole did not exceed 5. The child was presented with the sum orally and was asked, ‘‘How much is…?” There were a total of five items (α = 0.95).

**First Grade**. The task included 30 simple sums with numbers between 1 and 16. The children were asked to give correct answers to as many addition problems as possible within a limit of 120 s (α = 0.89).

**Verbal Story Problems.** Kindergarten. The child was presented verbally with five basic addition (α = 0.57) and five subtraction (α = 0.64) problems. Each item was read to the child, who then was asked to solve the problem by stating a number word verbally.

**First Grade**. Five basic addition verbal story problems (α = 0.53) and five subtraction (α = 0.58) problems of the same type were read out loud to the child, who then was asked to solve the problem by stating a number word orally.

#### 2.2.2. Early Literacy

**Word Recognition**. This is the Hebrew version of a Dutch test adapted from [58]. The examiner presented a list of four real words. Each list included a target word and three distractors that differed in one, two, or all letters from the target word. The child was asked to identify the target word said out loud by the examiner. The test included 12 items which were scored from 3 to 0 points (α = 0.77).

**Letter Naming.** The child was asked to name 10 Hebrew letters presented separately on A4 paper. The total of letters correctly named was calculated (α = 0.87).

**Letter Identification.** The test includes a total of 10 items. The child had to identify a target letter from four letters presented. The total of correct letters correctly identified was calculated (α = 0.82).

**Phonological Awareness.** Three different phonological-awareness tasks were administered: two initial consonant isolation tasks in CVC (α = 0.81) and CCVC syllables (α = 0.84), and a final phoneme isolation task in CVC syllables (α = 0.89). The words in each task were presented orally, and the child was required to say the initial or the final consonant syllable out loud. Each test was preceded by three practice items in which detailed feedback was provided by the examiner. Testing was concluded once a child made five sequential errors. The total of correct answers was calculated separately for the initial (CVC and CCVC syllables) and final consonant-isolation tasks (maximum = 10).

**Vocabulary.** The picture-naming task was based on the vocabulary subtest from a language-screening test for Hebrew-speaking kindergarten children. The test contains 14 color pictures printed separately on an A4 paper. The children were asked to name each picture out loud following the examiner’s instructions (e.g., “What is this?”, “What is he doing?”). The test was preceded by one practice item. The score was based on the total pictures named correctly (α = 0.75).

**Morpho-Syntactic Skills.** Nonword derivation task. The test includes 10 sentences that were presented orally by the examiner. Each sentence contains a novel verb (a combination of root and conjugation) which represents a nonsense word in the Hebrew language. The children were required to complete the sentences by modifying and producing the verb in the correct inflection and derivation according to Hebrew morpho-syntactic structures and rules (α = 0.65).

**Noun Plural Production.** Children were shown a piece of A4 paper containing two colored pictures. One picture showed a singular count noun item and the second showed four exemplars of that same item. The examiner said a sentence while pointing to the single count noun item, for example, “Here is a cherry”. Following this, the examiner pointed to the second picture (with four cherries in it) and said, “These are a lot of…” with rising intonation to encourage the student to complete the sentence with the plural. The total of correct answers was calculated (maximum = 15, α = 0.74).

**Consequential Adjective Production.** The test contains 10 items printed on pieces of A4 paper. Two colored pictures were displayed on each page. While pointing to the first picture, the examiner said a sentence with a target verb, for example: “They **broke** the window” (the verb Shaveru “broke”). Following this, the examiner pointed to the second picture (with a **broken** window) and asked the children to complete the sentence by deriving the consequential adjectives from the verb: “Now the window is…?” (Shavur “broken”) The test was preceded by two practice items (α = 0.74).

**Consequential Verb Production.** The test includes a total of eight sentences read out loud by the examiner. The children were required to complete the sentence by deriving the consequential verb from a noun. For example: “What are we doing with the paint?” (Tzeva) “With the paint, we are…” (Tsovei’em “painting”). The total number of correct answers was calculated (α = 0.74).

**First Grade Reading Test.** This test is based on the TOWRE Test, which is based on [59]. The test assesses reading fluency. In this task, participants were asked to read aloud 80 single words as quickly and as accurately as possible in 45 s. The words were ordered by increasing difficulty. The reliability of this task is α = 0.95. Scores were calculated according to the number of correct words provided in 45 s.

#### 2.2.3. Simple Working Memory Skills

##### Verbal Auditory Memory Tests

**Digits Forward** (WISC-3R; [60]). This test examined short-term auditory memory (α = 0.67). The child was asked to repeat a series of numbers said aloud by the examiner in ascending order of difficulty—first a series of two numbers, and if she managed to remember them, the number of items in each series was increased. The score consisted of the number of digits the child remembered correctly

**Word Repetition Test**. (Based on [61]). This test examined short-term auditory memory (α = 0.61). This task tested children’s ability to repeat a sequence of words in the order the word were read to them. The test began with a series of two words. If the child succeeded, in the next stage another word was added to the series. The score was calculated according to the number of words in sequence correctly remembered by the child

##### Auditory-Visual Memory

**Word Order Test** [62]. Kindergarten. This test is designed to examine the integration between auditory memory and visual memory. The task tested children’s ability to point out a sequence of images that displayed objects whose names were read out to them. The test began with a series of two words. If the child succeeded, in the next stage another word was added to the series. The score was calculated according to the number of words in sequence correctly remembered by the child (α = 0.75).

##### Visual Spatial Memory

**Spatial Memory—Matrices** [62]. This test examined short-term auditory memory (α = 0.86). This test examines short-term visual spatial memory. It checks the child’s ability to repeat a sequence of illustrations by the location in which they appeared on a matrix of squares. The test began with a series of two stimuli in a nine-square matrix. If the child succeeded, in the next stage another illustration was added to the matrix. The score consisted of the number of items correctly remembered by the child.

**Spatial Sequential Memory—Corsi Frog**, adapted from DEST-2 [63]. This test examined short-term auditory memory (α = 0.81). Materials for the task included a colored, printed card with seven lily pads arranged in a random order along with a small plastic toy frog. The children were required to watch the frog make a series of jumps on the lily pads and then to copy the frog’s jumps in the same order. A practice session was administered, followed by instructive feedback. The difficulty of the tasks gradually increased, starting with blocks of two jumps and gradually increasing to blocks of seven jumps. The capacity score was calculated based on the longest-length list correctly recalled.

**Hand Movement Test** [62]. Kindergarten. This test examined children’s ability to repeat a sequence of hand movements as displayed to them with an increasing level of difficulty. The test began with a series of two hand movements. If the child succeeded, in the next stage another movement was added to the series. The score was calculated according to the number of movement sequences correctly remembered by the child (α = 0.86).

#### 2.2.4. Complex WM Skills

##### Complex Visual Spatial WM

**Corsi Frog Backward**, adapted from DEST-2 [63]. This test examined short-term visual memory (α = 0.81). This is similar to the forward test, but in this version the children had to copy the frog’s jumps in the reverse order.

**Series of Numbers** [64]. This test examined working visual memory (α = 0.68). The test examined the children’s ability to retain data (number sequences) while simultaneously naming colors. The children were presented with a number (digits from 1–9). Following each display of numbers, a randomly dotted page was presented; one of the dots was colored and the rest were black. In each group of dots, the colored dot was of a different color. The children were asked to name the color of the dot while simultaneously remembering the sequence of digits presented to them. At the end of each stage of the test, children were asked to repeat the numbers they viewed in the correct order.

##### Complex Verbal Auditory WM

**CSOT** [65]. This test examined short-term auditory memory (α = 0.72). In this test, the examiner read a list of words aloud to the child. At first, two words were read, and if the child succeeded in remembering them, the number of words was increased. Then, the child was asked to say the words by size, from the word that represents the smallest item to the word that represents the largest item. For instance, the examiner said: “Lion, doll”, and the child was supposed to respond: “Doll, lion”. For a wrong answer the child received no points, for self-correction one point, and for a correct answer two points.

**Digit Span Backward test** (WISC-III, [60]). This test examined short-term auditory memory (α = 0.72). This is the same as the digit-span forward test, but in this version the children were instructed to repeat the digits in reverse order, from the last number they had heard to the first one.

### 2.3. Procedure

Prior to the collection of the data, the required approvals were obtained from the Ministry of Education as well as the ethics committee of the relevant university. In addition, consent forms were signed by the parents of the children examined. All the tests were administered to the participants individually during kindergarten time but in a separate room, in two or three separate sessions of about 20 min each. In the first grade, the children were also tested individually in a separate room in two sessions of 40 min. All tests were administered in a random order.

### 2.4. Data Analysis

In order to test our hypothesis about the relationships among the components of WM and early literacy and early numeracy in kindergarten and math and language skills in first grade, we used structural equation modelling (SEM) by AMOS (Analysis of Moment Structures). We defined a measurement model for 11 latent variables; the concepts were measured by single indicators. For all indicators, an independent measurement error was estimated.

The latent variables were constructed based on Baddeley’s multicomponent WM model (1974, 2000), and prior research in language and math fields [12,15,16]. In the field of WM, a division was made into simple WM and complex WM, where a subdivision of simple WM was verbal auditory WM (VASimple WM) and visual WM (VISimple WM). The tests conducted were sorted according to the above division. In the field of early literacy and early numeracy, the division of assignments and tests was undertaken according to the tested skills. The indicators were carefully selected based on their theoretical relevance and ability to represent different aspects of the latent variable. This theoretical framework served as the foundation for our research, providing a roadmap to navigate the complexities of the latent variables. As we analyzed the data, we refined and iterated upon the measurement model, ensuring that the selected indicators adequately captured the latent variable. We assessed their reliability and validity, seeking to establish a robust measurement structure that aligned with the theoretical expectations.

Details of the variables and correlations can be found in the tables in the appendix.

Eight latent variable (LV) measurement models were created, corresponding to Working Memory: Simple Verbal Auditory WM (VASimple WM); Simple Visual Spatial WM (VISimple WM) and Complex WM; Early Literacy and Language Skills: Phonological Awareness (PHON), Orthographic Knowledge (ORTH) Morphological Knowledge and Vocabulary (MORPH), and Reading; Early Numeracy and Math Skills: Relations (Rel), Counting, Arithmetic Skills (Arithm) and Number Sense (Num sense). The subject of the investigation was the strength of the relationships between the WM and language latent variables (LVs) and the relationships between the WM and math LVs. The fit of the data to the model was tested by the structural model. For each latent variable, an independent measurement error was estimated. The parameters were estimated based on all available data.

The models’ fit with the data was compared in two steps. First, the models were fitted separately to each sample with maximum likelihood estimation. Maximum likelihood estimation has been shown to outperform most common methods of handling missing data, including mean substitution [66]. Second, the models’ fit to the data was evaluated using the following goodness-of-fit indices: chi-square ($\chi^{2}$), root mean square error of approximation (RMSEA), Tucker–Lewis index (TLI), and comparative fit indices (CFI). RMSEA is usually accepted for values below 0.07; for TLI and CFI fit indices are usually considered acceptable for scores equal or higher than 0.90. A generally accepted standard of the goodness-of-fit of the $\chi^{2}$ statistic is in comparison to the degrees of freedom where a value of between one and three is reported to be an acceptable fit. However, the $\chi^{2}$ measure is sensitive to sample size; therefore, the value of the $\chi^{2}$ as a measure of goodness-of-fit is ambiguous [66].

## 3. Results

### 3.1. Measurement Models

#### 3.1.1. Early Literacy and Language Skills with Memory Abilities

CFA models were conducted based on phonological awareness, orthographic knowledge, morphological knowledge, and vocabulary variable clusters in kindergarten, with addition of reading skills in first grade. CFA models were conducted based on simple verbal auditory WM (VASimple WM) and simple visual spatial WM (VISimple WM) as well as complex WM. All CFAs produced adequate fit to the data (see Table 1) (i.e., CFIs exceeding 0.90, and RMSEA less than or equal to 0.07).

The overall measurement models, including all language and simple WM or complex WM latent variables, also produced satisfactory model fit, indicating that the observed variables were good indicators of the latent variables and the latent variables represented separate constructs.

#### 3.1.2. Early Numeracy and Mathematical Skills with Memory Abilities

CFA models were conducted on relations, counting, arithmetic skills, and number sense variable clusters in kindergarten and in first grade.

The overall measurement models, including all math and simple WM or complex WM latent variables, also produced satisfactory model fit (see Table 1), indicating that the observed variables were good indicators of the latent variables and that the three latent variables represented separate constructs.

### 3.2. Structural Models

Early Literacy in Kindergarten and Language Skills in First Grade

Simple WM. The models of relationships between simple WM and early literacy in kindergarten fit to the data: $\chi^{2}$ (180, *N* = 250) = 368.822, *p* < 0.001; CFI = 0.91; TLI = 0.91; RMSEA = 0.07.

The coefficients from VASimple WM to phonology knowledge, β = 0.38, *p* < 0.05, and to morphology and vocabulary β = 0.89, *p* < 0.001 (see Figure 1), indicate that VASimple WM was significant only in relation to verbal components of early literacy, when the strongest connection was to morphology and vocabulary. VASimple WM was not significantly related to orthography knowledge. The coefficients from VISimple WM to orthography knowledge were β = 0.35, *p* < 0.05. VISimple WM seems to have a medium and meaningful connection with orthography knowledge; however, no contribution of VIsimple WM to other types of early literacy abilities was found.

The models of relationships between simple WM and language skills in first grade fit to the data as follows: $\chi^{2}$ (123, N = 137) = 173.890, *p* < 0.01; CFI = 0.94; TLI = 0.94; RMSEA = 0.06. The coefficients from VASimple WM to phonology knowledge are β = 0.78, *p* < 0.001; to morphology and vocabulary β = 0.85, *p* < 0.001; to orthography knowledge β = 0.75, *p* < 0.01; and to reading β = 0.39, *p* < 0.05 (see Figure 2), indicating that VASimple WM was significantly related to all components of language skills, when the weakest connection was to reading.

The coefficients from VISimple WM to reading were β = 0.51, *p* < 0.05. VISimple WM seems to have a medium and meaningful connection to reading; however, no contribution of VISimple WM to other types of language abilities was found.

Simple visual and verbal memory abilities among kindergarteners explained 28% of the variance in phonology, 23% of the variance in orthography, and 40% of variance in morphology and vocabulary. Simple visual and verbal memory abilities in first grade explained 60% of the variance in phonology, 65% of variance in orthography, 74% of variance in morphology and vocabulary and 41% of variance in reading.

### 3.3. Complex WM

The models of relationships between complex WM and early literacy fit to the data: $\chi^{2}$ (146, *N* = 250) = 349.003, *p* < 0.001; CFI = 0.90; TLI = 0.90; RMSEA = 0.07.

The coefficients from complex WM to phonology knowledge, β = 0.52, *p* < 0.001, to orthography knowledge β = 0.43, *p* < 0.001 and to morphology and vocabulary β = 0.44, *p* < 0.001 (see Figure 3), indicate that complex WM was significantly related to all components of early literacy, and the strongest connection was to phonology.

The models of relationships between complex WM and language skills fit to the data: $\chi^{2}$ (124, *N* = 137) = 187.787, *p* < 0.001; CFI = 0.92; TLI = 0.93; RMSEA = 0.06.

The coefficients from complex WM to phonology knowledge are β = 0.71, *p* < 0.001, to orthography knowledge β = 0.87, *p* < 0.001, to morphology and vocabulary β = 0.68, *p* < 0.001 and reading β = 0.67, *p* < 0.001 (see Figure 4), indicate that complex WM was significantly related to all components of early literacy, and the strongest connection was to orthography.

Complex WM memory abilities in kindergarten explained 27% of the variance in phonology, 19% of variance in orthography, and 20% of variance in morphology and vocabulary, whereas in first grade the same abilities explained 51% of the variance in phonology, 75% of the variance in orthography, 46% of the variance in morphology and vocabulary, and 45% of the variance in reading.

#### 3.3.1. Early Numeracy and Mathematics Skills

##### Simple WM

The models of relationships between simple WM and early numeracy fit to the data, $\chi^{2}$ (282, *N* = 250) = 497.046, *p* < 0.001; CFI = 0.95; TLI = 0.95; RMSEA = 0.05.

The coefficients from VASimple WM to relations, β = 0.57, *p* < 0.001, to counting, β = 0.47, *p* < 0.001, to arithmetic skills, β = 0.67, *p* < 0.001 and to number sense, β = 0.49, *p* < 0.001 (see Figure 5) indicate that VASimple WM was significantly related to all components of early numeracy, when the strongest connection was to arithmetic skills.

The coefficients from VISimple WM to relations, β = 0.56, *p* < 0.01, to counting, β = 0.46, *p* < 0.001, to arithmetic skills, β = 0.29, *p* < 0.01, and to number sense, β = 0.67, *p* < 0.001 (see Figure 5) indicate that VISimple WM was significantly related to all components of early numeracy, when the strongest connection was to number sense. The models of relationships between simple WM and mathematics skills fit the data: $\chi^{2}$ (157, *N* = 137) = 210.731, *p* < 0.01; CFI = 0.93; TLI = 0.95; RMSEA = 0.05.

The coefficients from VASimple WM to relations, β = 0.73, *p* < 0.05, to counting, β = 0.69, *p* < 0.01, and to arithmetic skills, β = 0.67, *p* < 0.001 (see Figure 6) indicate that VASimple WM was significantly related to three components of mathematics skills, when the strongest connection was to relations. However, VASimple WM was not found to contribute to number sense.

The coefficients from VISimple WM to relations, β = 0.51, *p* < 0.05, to arithmetic skills, β = 0.46, *p* < 0.05 and to number sense, β = 0.83, *p* < 0.05 (see Figure 6) indicate that simple visual spatial WM was significantly related to three components of mathematics skills, when the strongest connection was to number sense. However, no contribution of VISimple WM to counting was found.

Simple visual and verbal memory abilities in kindergarten explained 47% of the variance in counting, 64% of the variance in relations, 54% of arithmetic skills, and 69% of the variance in number sense. In first grade, the same memory abilities explained 54% of the variance in counting, 78% of the variance in relations, 66% of arithmetic skills and 78% of the variance in number sense.

##### Complex WM

The models of relationships between complex WM and early numeracy fit to the data: $\chi^{2}$ (239, *N* = 250) = 445.866, *p* < 0.001; CFI = 0.90; TLI = 0.90; RMSEA = 0.06.

The coefficients from complex WM to relations, β = 0.54, *p* < 0.01, to arithmetic skills, β = 0.61, *p* < 0.001, to counting, β = 0.59, *p* < 0.001 and to number sense, β = 0.63, *p* < 0.001 (see Figure 7), indicate that complex WM was significantly related to all components of early numeracy, when the strongest connection was to number sense.

The models of relationships between complex WM and mathematics skills fit the data: $\chi^{2}$ (163, N = 137) = 230.149, *p* < 0.001; CFI = 0.92; TLI = 0.92; RMSEA = 0.06. The coefficients from complex WM to relations, β = 0.89, *p* < 0.01, to arithmetic skills, β = 0.87, *p* < 0.001, to counting, β = 0.86, *p* < 0.01 and to number sense, β = 0.70, *p* < 0.01 (see Figure 8), indicate that complex WM was significantly related to **all** components of early numeracy, when the strongest connection was to relations.

Complex WM memory abilities among kindergarteners explained 37% of the variance in counting, 27% of the variance in relations, 37% of arithmetic skills and 40% of variance in number sense. In first grade, the same memory abilities explained 74% of the variance in counting, 81% of the variance in relations, 76% of the variance in arithmetic skills, and 49% of variance in number sense.

In order to summarize our findings, we made two comparison tables. Table 2, which gives an overview of the results for the language domain, shows that VASimple WM was significantly related to all components of language skills in first grade and most components of early literacy in kindergarten. However, the strongest connection was to orthographic knowledge and phonological awareness in first grade, whereas no contribution of this component of WM to orthographic knowledge in kindergarten was found. The contribution of VASimple WM to morphological knowledge and vocabulary in first grade and kindergarten seems to be the same.

VISimple WM was found to be related only to reading in the first grade and to orthographic knowledge in kindergarten. Complex WM was significantly related to all components of early literacy in kindergarten and to all language skills in the first grade, and its contribution is greater in first grade. In order to examine the differences between the different correlations, a Fisher test was conducted and the significance of the results is presented (see Table 2). In the mathematics domain, VASimple WM was significantly related to all components of early numeracy in kindergarten and most components of mathematical skills in first grade (see Table 3). However, the strongest connection was to relations and counting in first grade, whereas this component of WM was not found to contribute to number sense in first grade. VISimple WM was found to be related to all components of early numeracy and most components of mathematical skills in the first grade. The strongest connection was to relations and counting in kindergarten and to arithmetic skills and number sense in first grade. This component of WM too was not found to contribute to counting in first grade. Complex WM was significantly related to all components of early numeracy in kindergarten and mathematical skills in the first grade, while the strongest connection was to all math skills in the first grade. In order to examine the differences between the different correlations, a Fisher test was conducted and the significance of the results is presented (see Table 3).

Figure 9 below describes the contribution of complex and simple WM to variance in academic achievements of kindergarteners and first graders. It seems that the influence of all components of WM is stronger on first graders. All the simple WM skills explained between 23% and 40% of the variance in early literacy, and between 47% and 69% of early numeracy. In first grade, all the simple WM skills explained between 41% and 74% of the variance in language skills, and between 54% and 78% of math skills. All the complex WM skills explained between 43% and 52% of the variance in early literacy and between 27% and 40% of the variance in early numeracy.

In the first grade, all the complex WM skills explained between 45% and 75% of the variance in language skills and between 49% and 81% of variance in math skills. In addition, it seems that complex WM contributes significantly mainly to early literacy in kindergarten and to math skills in the first grade. However, the influence of simple WM is stronger on early numeracy in kindergarten and on language skills in the first grade.

## 4. Discussion

A large number of studies have pointed out the role of WM throughout numerical and language development (see [25], for a review; e.g., [67,68,69,70]. Many studies have tried to explain the contribution of the specific elements of WM to skill-acquisition processes such as reading and mathematical skills (e.g., [30,70,71,72]), but there is no definitive agreement regarding the influence of memory components on early literacy and early numeracy among kindergarten-age children and math and language skills among first graders.

The purpose of our study was to investigate the contribution of the three WM systems to specific components of early literacy and early numeracy and to math and language skills one year later. The statistical models which are presented in this study analyze the influence of various WM components on acquiring different academic skills. Consistent with our predictions and in line with the findings of other studies (e.g., [31,73,74]), we found strong evidence for the significant relations between all components of memory and early mathematics and language knowledge, with some differences in the developmental process from kindergarten to first grade.

### 4.1. Contribution of the Simple Memory System to Numeracy and Language Skills

The simple WM (verbal auditory and visual spatial) abilities contributed significantly to both mathematical and linguistic abilities in kindergarten and first grade. Based on the findings, it is possible to conclude that there is a greater contribution to mathematical knowledge than to language knowledge; furthermore, its contribution is higher in the first grade in both math and language skills. The most notable contribution to the language domain both among kindergarteners and first graders was found in morphological knowledge and vocabulary, concurrently with the contribution to math skills in number sense and relations.

Our findings are consistent with theoretical framework of Baddeley’s multicomponent WM model (1974, 2000) and lend support to previous evidence that revealed the key role of specific components of WM in the emergence of initial literacy and the development of number sense. Studies with kindergarteners have shown the relationship between phonological or visual spatial memory and book and print awareness, phonological awareness, vocabulary development, the development of reading comprehension, and early reading [75,76]. In the field of early numeracy, similar to our findings, other studies have suggested that verbal auditory and visual spatial memory develop differently and affect number word knowledge, counting, symbolic and non-symbolic number processing, problem solving, and calculation performance, among others [28,38,70,74]. Some studies suggest an increasing involvement of verbal WM skills in mathematical and language cognition as children grow older [43,77] (see [38] for a meta-analysis), while other researchers have pointed to the increasing contribution of the visual WM skills (e.g., [23,70,77]). The growing contribution of simple WM from kindergarten to first grade is reflected in our study findings.

In a specific observation of the different types of simple WM, significant connections were found between simple verbal auditory WM (VASimple WM) and a variety of skill areas in the language domain, including morphological knowledge and vocabulary and phonological awareness, both among kindergarteners and first graders, and to orthographic knowledge and reading in first grade. In the math domain, the strongest connections were found with counting, relations, and arithmetic skills, both among kindergarteners and first graders.

As far as early literacy is concerned, VASimple WM appears to be involved in the acquisition process of all linguistic skills in the first grade. Accordingly, the findings of many studies have pointed to the role of the phonological memory in a broad set of phonological processing skills, such as identifying and manipulating units of oral language parts (e.g., words, syllables, onsets and rimes, and phonemes) (e.g., [78]). Studies of school-aged children have also found that vocabulary knowledge was strongly related to verbal short-term storage [79,80] and suggested that VASimple WM is the driving force behind the formation of stable phonological representations of new words in long-term memory and underpins vocabulary development.

The contribution of VASimple WM to orthographic knowledge and reading in first grade could be explained by the relationship between phonology and the writing system. Phonology is at the core of reading and writing; every writing system is based on the specific letters and their sounds as well as meaning (morphology and semantics). Reading cannot be achieved without the ability to decode new words, and this can only be accomplished via a workable degree of phonological transparency—systematic or semi-systematic correspondences between symbols and sound [15].

Despite the disagreement that exists in studies regarding the contribution of verbal memory to mathematical development, some of our findings in the field of *early numeracy* have support in the research literature. In particular, regarding the relation of VASimple WM to basic counting, relations, and arithmetic skills in our findings, other measures of the phonological loop (in our study—VASimple WM) have also been found to be correlated with counting, addition skills [74], and basic fact retrieval [5] (. Several studies have suggested that the phonological loop is used when children need to transform different symbols and number strings into spoken words, such as when using verbally mediated counting strategies while performing different arithmetic problems [30,61,74].

Although the research literature has found a link between a phonological loop (in our study—VASimple WM) and number sense [33], our study findings did not reflect this result. The difference in the findings could be due to peculiarities in the definition of the variable itself and its components and possibly because of the task requirements: both the input and the output of the tasks were primarily based on visual items, especially in first grade, and VASimple WM was probably less involved. Further research is needed to explore the factors that might influence the relationship between the VASimple WM and number sense in different contexts.

Even though the relationship between visual spatial WM (VISimple WM) and the development of language skills has been less studied, VISimple WM was found to be significantly related to orthographic knowledge in kindergarten and to reading in first grade. These relationships seem relevant and consistent with the findings of other studies that noted the contribution of VISimple WM to children’s achievements in the reading and writing domains at the start of formal schooling. The impact of this subtype of memory was found on reading comprehension among young children [52]; on children’s particular writing ability at five years of age [81]; and on the process of acquiring reading and writing skills, when the children are required to decode recognized visual stimuli, such as letters or symbols, and applying them into other situations [76].

The lack of a significant connection between VISimple WM and orthographic knowledge in first grade suggests that other cognitive factors or skills may be more influential in the development of orthographic abilities at that stage. It is possible that other factors, such as phonological processing, grapheme–phoneme correspondence, or vocabulary knowledge, may play a more prominent role in the development of orthographic knowledge in first grade. It is important to note that the absence of a significant relationship in this study does not mean that VISimple WM is irrelevant or unimportant for orthography in all contexts. Additional research is necessary to further explore the specific interactions between VISimple WM and orthographic skills.

In support of our findings regarding a link between VISimple WM and number sense, It has been demonstrated that symbolic magnitude processing correlates with visual spatial memory abilities [82]. In addition, the relation between VISimple WM and early numeracy as reflecting the visuospatial nature of numerical representations was also found [69].

Several studies detected the correlation between spatial skills and VISimple WM and children’s early counting and general mathematical competence (e.g., [83]), finding that the visuospatial sketchpad was the best predictor of early math learning [25,84]. It was also discovered that the visuospatial sketchpad was especially important to the development of mental arithmetic in young children (under the age of seven); in order to use their mental number line, they need to relay on the visuospatial encoding [85]. It has been found that children at this age rely mainly on visual memory to remember information, rather than converting the visual items to verbal labels, which relies on the phonological loop [85]. This assumption is partially reflected in our findings. In this study, VISimple WM contributed to most domains of math skills in both first and kindergarten grades; only in counting skills in first grade was there no contribution of this memory component. It is important to note that the contribution of VISimple WM was more significant in all tasks.

Our findings are inconsistent with De Vita and colleagues (2021) [4] who found that the phonological loop turns into a predictor of performance on verbal problems, as well as that nonverbal and verbal mathematical task performances are similar in first grade. Several researchers claim [70,85] that kindergarteners may solve some mathematical problems by the use of mental models, and, just as language skills develop as well as verbal memory, children may begin to rely more on verbal memory capacity in order to perform different mathematical tasks. Our findings could support the claim that different components of WM may have variant relevance for distinctive early numeracy and math skills in later years, because the WM demands of tasks with different properties may vary. Non-symbolic tasks may rely heavily on visuospatial skills [86], while the presentation format of the problem can involve various components of WM.

Previous studies have suggested that verbal and visual spatial WM resources seem to develop independently [70,87]; in the current study, we found different effects of the two components of simple WM as well. In addition, the present study showed the strength of association between verbal auditory and visual spatial short-term memory (simple WM) in kindergarten, but not in first grade, and that different tasks require different cognitive resources. This is in line with the view that the two variables increase slightly from the four- to six-year-old age group [88] and the two systems are separate. Performance in VASimple WM tasks would not predict spatial abilities, nor would VISimple WM measures be related to verbal skills. As a result, we can carefully assume that the difference in resources required may depend on the type of task and instructions given. This idea awaits confirmation in additional studies.

### 4.2. Contribution of the Complex WM System to Numeracy and Language Skills

We succeeded in exploring the contribution of separate components of the simple WM to early academic performance. However, we were unable to separate the verbal and visual components of the complex WM. Our findings are broadly consistent with other studies, in which the correlation between verbal and visual spatial WM was significantly high and the two variables shared a large amount of variance (83%) [88]. In addition, a study which trained memory abilities among kindergarteners showed that verbal and visual spatial systems of WM could not be distinguished as two separate systems in young children either [68].

Our findings are in line with Baddeley’s model that claims that the central executive (complex WM in our study) is the most important component, which controls the other components by coordinating visual and verbal information and balancing WM and long-term memory. The complex WM is not modality specific; it is also involved in selective attention, inhibiting or suppressing automatic responses, updating the WM with new information, and shifting between tasks [7].

An analysis of complex WM skills and their contribution to math and language skills shows that complex WM explains the variance in both of the academic domains, but with a higher level of variance in language than in mathematics in kindergarten, whereas in first grade the contribution of this memory component is higher in mathematic achievements as opposed to language. As with simple WM, the contribution of complex WM is higher at school age in both math and language skills. This finding is consistent with the findings that highly controlled WM processes are more strongly related to higher cognitive abilities [77].

In early literacy, the complex WM played a more significant role in phonological awareness, whereas in first grade, orthographic knowledge contributed more strongly. It is important to note that the contribution of complex WM is considerable in all linguistic components, especially in first grade. This finding on phonological awareness is in line with the theoretical framework suggesting that WM plays a significant role in the different phonological awareness tasks [72]. WM is needed for the complex processing and storage of these phonological representations (e.g., [89]), because of the complexity of different phonological coding tasks. The strong evidence for the influence of complex WM on orthographic knowledge and reading is in line with the suggestion that orthographic activity combines the processing and manipulation of visual information. As a result, WM is required for the simultaneous processing and storage of orthographic representations as well. Our finding corresponds with the suggestion that the complex WM is more critical for coordinating and integrating the different information which is processed from the visual words read [7].

It seems that morphological knowledge and vocabulary tasks also require complex WM resources. Evidence has also been found regarding the connection between WM and vocabulary (e.g., [90]). To perform a vocabulary task, complex WM may be used to simultaneously process verbal/visual information, activate relevant background knowledge and concepts, and integrate those sources of information [72]. Our results are in line with Noël’s (2009) [74] findings that at the age of four or five, measures of complex WM are better predictors of vocabulary than measures of VASimple WM, and fit with theoretical views that the complex WM is a system that is able to encode and retrieve information from the short term as well as the long-term memory [7].

In the math domain, the complex WM was related with all sub-fields of mathematical knowledge in both age groups. This finding is supported by the research literature. Many studies have emphasized the role of complex WM throughout numerical development (see [25] for a review). Several studies have shown that the complex WM abilities of young children (five to seven years old) are correlated with mathematical performance such as counting [40] or number recognition, which involves linear representations of number [32]. Children who are poor in mathematics also perform poorly in central executive tasks [91,92,93]), while weak complex WM is associated with difficulties in early numeracy [94,95], with slower learning of the counting strings [74] and lower accuracy in solving simple addition problems [37,74]. Complex WM processes contribute to developing new semantic representations and to growing foundational mathematical skills [96].

Noël’s (2009) [74] conclusions also lend support to our findings: in her research, complex WM was the best predictor of all the tasks which included numerical vocabulary and counting, and the link with complex WM was even stronger than the link with VASimple WM. Studies on school-age children indicate that mathematics performance is most strongly correlated with complex WM tasks (e.g., [77]). It is important to note that the contribution of the complex WM is significant in all math components, especially in first grade.

Our study findings in kindergarten are consistent with previous research [38,97,98] indicating that variance in non-symbolic processing was explained by complex WM. However, in first grade, number sense processing should be more intuitive and require less WM involvement [75]; this point can be examined in future studies.

Our finding of a significant contribution of complex WM to relations, counting, and arithmetic skills in first grade is in line with previous studies. Earlier research has suggested that the significant associations between symbolic processing and WM components can also be explained by the fact that WM components are highly involved in the development of these skills [75] and that the significant associations between symbolic processing and both the central executive and visuospatial sketchpad contribute to successful counting and use of number words and symbols [38]. Previous studies have found that even simple mathematics calculations and number-comparison tasks involve WM processes such as the temporary storage of problem information, retrieval of relevant procedures, and processing operations to convert the information into numerical output [40].

During the school years, more complex arithmetic procedures involve more complex abilities which demand integration and retrieval of different information. Therefore, complex WM plays a major role across all types of single- and multi-digit arithmetic operations, because of the split in attentional resources and the necessity of maintaining intermediate results (for reviews, see [25]).

## 5. Conclusions

All the WM components seem to be involved in mathematics and language processes, but they have somewhat different roles. The impact of simple WM on the learning of math and language skills is critical in early stages of the acquisition and retention of foundational knowledge in these domains. The complex WM seems to play a pivotal role across all types of tasks, while the connection of complex WM was stronger in its contribution and more significant in both mathematics and language domains in first grade. Complex WM resources become a more important information-processing tool or resource at kindergarten age in early literacy, whereas the simple WM seems to be important in early numeracy. Additionally, the contribution of the simple WM in first grade is stronger in the math domain.

In line with Noel’s approach, we suggest that children’s performance on academic tasks is constrained by the WM involved in the specific situation and therefore is dependent on the type of task and stimuli presented. This suggestion awaits confirmation in additional studies.

Although it is empirically impossible to separate complex WM functions in this age group, it is important to note that, based on our data, WM models that separate WM resources by modality into visual spatial and verbal ones, such as Baddeley’s (2000), seem to be an important starting point when furthering our understanding of children’s WM skills and their significance in learning.

### 5.1. Educational Impact and Implications

Evidence for the role of the three components of WM in early numeracy and early literacy in kindergarten and in academic achievements one year later is important for both cognitive theory and educational practice. Theoretically, the understanding of memory processes that are involved in children’s mathematics and language performance advances our understanding of how children learn. All of the WM components have been associated with the early acquisition of academic skills, but the contribution of each of them is different. The importance of several components of WM and the different weight given to each in mathematics and language learning may deepen our understanding of the cognitive mechanisms involved in mathematical and language development. Our findings highlight the importance of understanding the mechanisms underlying the associations between WM and math or language abilities and contribute to ongoing debates about whether WM is domain-general or domain-specific.

Children with impairments in WM often fail in classroom learning activities that place heavy demands on WM [28,47]. Classrooms and teaching methods designed to help compensate for this burden or avoid WM-related learning failures should be one key element of programs of educational support for learners [99]. Given the significant WM demands for specific skills, it may be important to account for children’s WM skills in the instruction of these specific domains by either modifying classroom curricula or individualized instructional activities: reducing task demands, including instructional aids, or scaffolding; presenting worked examples and ‘fading’ the problem-solving steps, i.e., removing them successively; or using teaching strategies [99] involving giving less information that must be processed and stored, making sentences shorter and less grammatically complex, and including only high frequency words [100]. All this may serve to minimize WM loads and the impact of WM deficits during learning.

### 5.2. Limitations and Implications for Future Studies

We note the following limitations. First, we focused exclusively on samples that included typically developing participants. However, disabilities are rather heterogeneous, and different types of disabilities may have different cognitive or skill-deficit profiles that could influence the impact of WM to a different degree [8,28,67,101,102]. Future studies are needed to specifically investigate the relation between early literacy and early numeracy and WM among specific disability groups. In addition, there are important variables, such as the effects of schooling, home environment, or other sources of individual differences that have not yet been taken into account.

It was empirically impossible to separate complex WM functions in this age group. One of the reasons for this could be that some of the tasks that tested complex visual WM required verbal output and were probably not completely “clean”. In future studies, the more accurate separation of skills can help with creating WM models that separate WM resources by modality into visual spatial and verbal ones. Furthermore, it is important to follow these children over time and examine the contribution of the different WM components to the development of the different academic skills.

Future research could explore the bidirectional relationship between WM skills and numeracy and literacy abilities. While the existing literature suggests that WM skills contribute to numeracy and literacy development, it is equally important to investigate whether engaging in numerical and language activities can enhance WM capabilities. Identifying the bidirectional nature of this relationship can have important implications for instructional approaches and interventions aimed at enhancing both cognitive and academic skills.

## Figures and Tables

**Figure 1 children-10-01285-f001:**
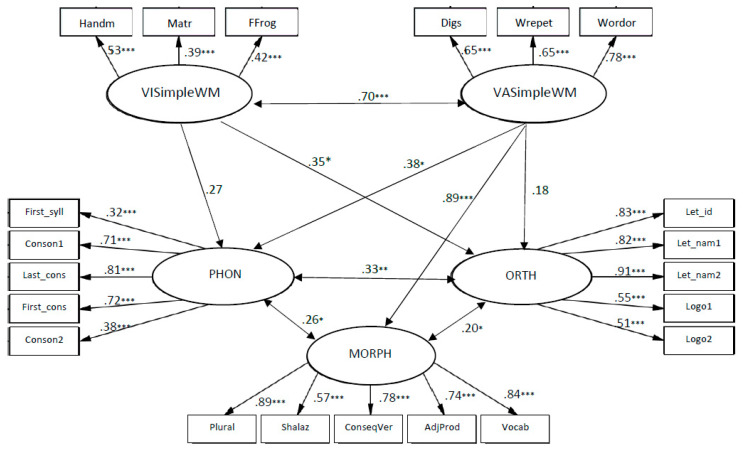
The relationship between simple WM and early literacy in kindergarten. * *p* < 0.05 ** *p* < 0.01 *** *p* < 0.001.

**Figure 2 children-10-01285-f002:**
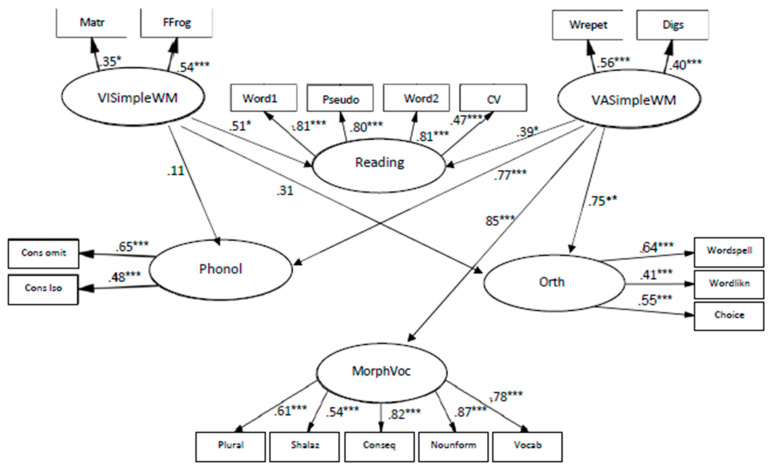
The relationship between simple WM and language skills in first grade. * *p* < 0.05 ** *p* < 0.01 *** *p* < 0.001.

**Figure 3 children-10-01285-f003:**
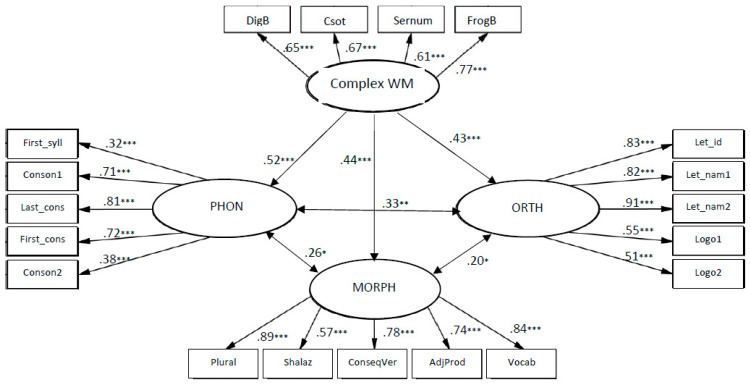
The relationship between complex WM and early literacy. * *p* < 0.05 ** *p* < 0.01 *** *p* < 0.001.

**Figure 4 children-10-01285-f004:**
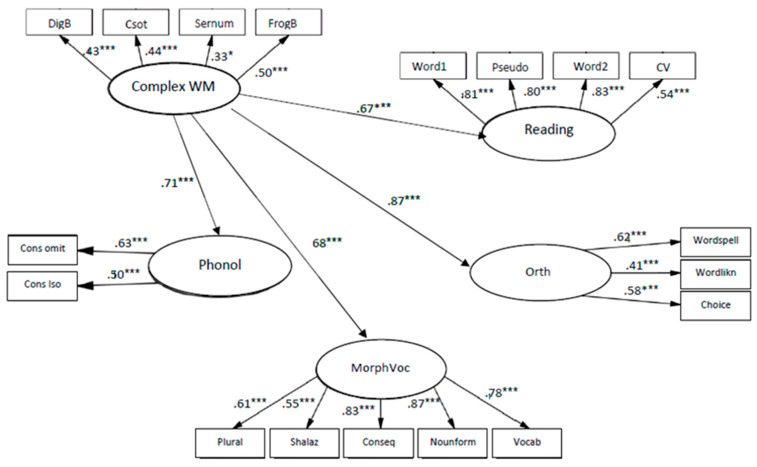
The relationship between complex WM and language skills. * *p* < 0.05 *** *p* < 0.001.

**Figure 5 children-10-01285-f005:**
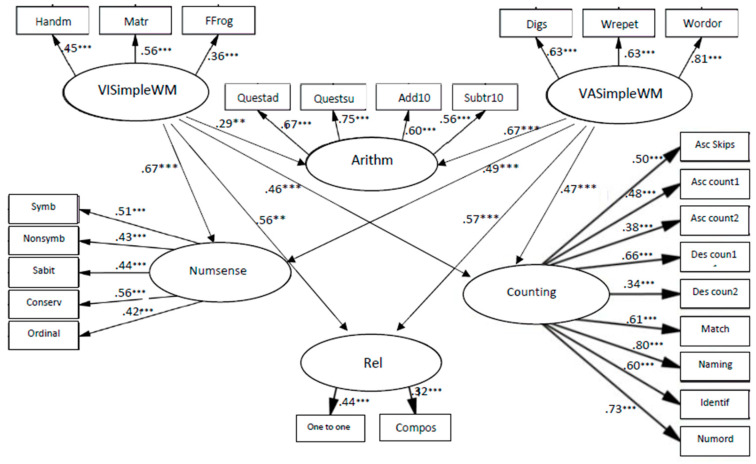
The relationship between simple WM and early numeracy. ** *p* < 0.01 *** *p* < 0.001.

**Figure 6 children-10-01285-f006:**
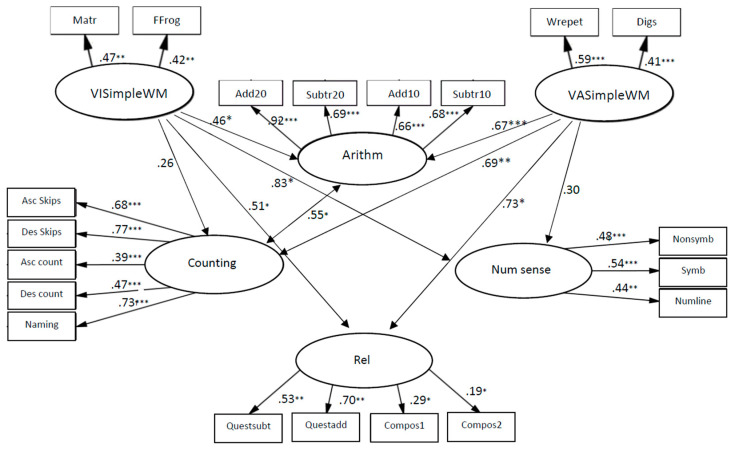
The relationship between simple WM and mathematics skills. * *p* < 0.05 ** *p* < 0.01 *** *p* < 0.001.

**Figure 7 children-10-01285-f007:**
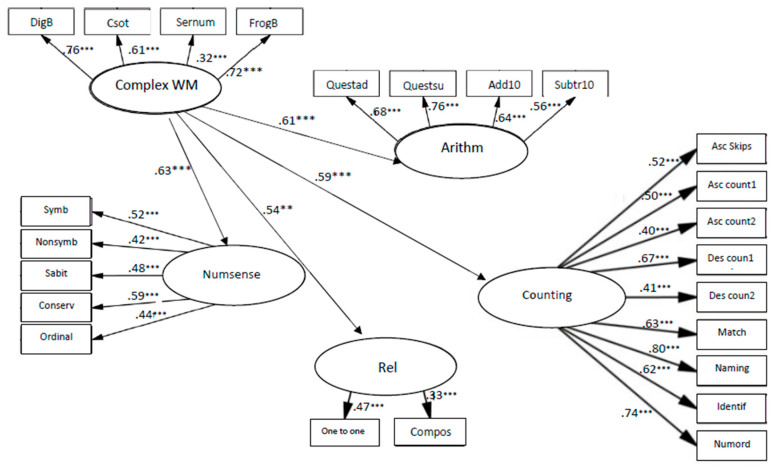
The relationship between complex WM and early numeracy. ** *p* < 0.01 *** *p* < 0.001.

**Figure 8 children-10-01285-f008:**
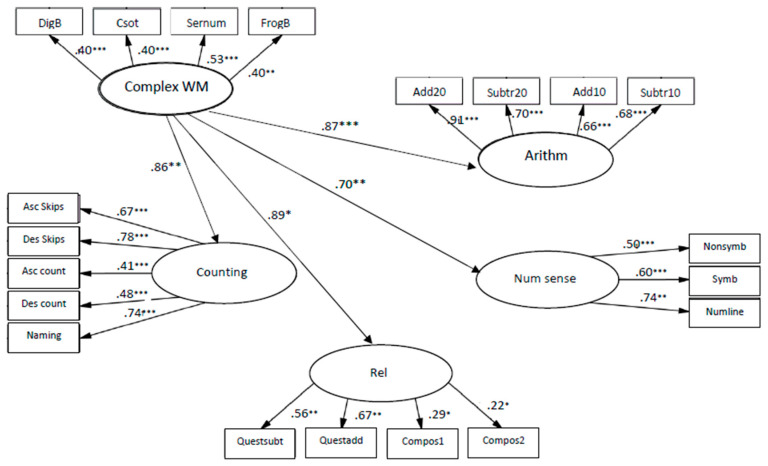
The relationship between complex WM and mathematics skills. * *p* < 0.05 ** *p* < 0.01 *** *p* < 0.001.

**Figure 9 children-10-01285-f009:**
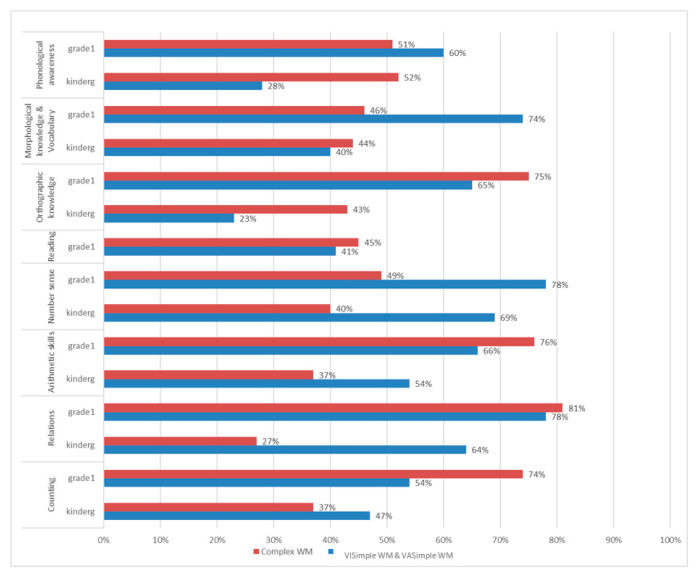
The Contribution of the Memory Measures to Variance in Language and Math Skills in Kindergarten and in First Grade.

**Table 1 children-10-01285-t001:** Model Comparisons.

	$\chi^{2}$ (Chi-Square)	Df (Dgrees of Freedom)	P (Significance)	$\chi^{2}/\mathrm{df}$ (Chi-Square Degrees of Freedom)	TLI (Tucker–Lewis Index)	CFI (Comparative Fit Indices)	RMSEA (Root Mean Square Error of Approximation)
Model 1: Simple WM and Early Literacy	348.071	178	0.000	1.955	0.922	0.920	0.062
Model 2: Complex WM and Early Literacy	294.870	143	0.000	2.062	0.928	0.927	0.065
Model 3: Simple WM and Language Skills	193.604	121	0.000	1.600	0.915	0.911	0.066
Model 4: Complex WM and Language Skills	201.700	123	0.000	1.640	0.909	0.906	0.069
Model 5: Simple WM and Early Numeracy	497.046	282	0.000	1.763	0.900	0.900	0.055
Model 6: Complex WM and Early Literacy	424.761	238	0.000	1.785	0.910	0.909	0.056
Model 7: Simple WM and Math Skills	208.802	153	0.002	1.365	0.931	0.928	0.052
Model 8: Complex WM and Math Skills	217.889	157	0.001	1.388	0.928	0.925	0.053
Kindergarten N = 250, First Grade N = 137 WM-Working Memory							

**Table 2 children-10-01285-t002:** Comparison between Kindergarten and First Grade Regarding the Relationship Involving WM Components and Language Skills.

	Reading	Orthographic Knowledge		Morphological Knowledge and Vocabulary		Phonological Awareness	
	Grade1	Kindergarten	Grade1	Kindergarten	Grade1	Kindergarten	Grade1
Simple Verbal Auditory WM	0.39 *	-	0.75 **	0.89 ***	0.85 ***	0.38 *	0.77 ***
Fisher correlation		*p* < 0.01		*p* = 0.05		*p* < 0.01	
Simple Visual Spatial WM	0.51 *	0.35 *	-	-	-	-	-
Fisher correlation		*p* < 0.01					
Complex WM	0.67 ***	0.43 ***	0.87 ***	0.44 ***	0.68 ***	0.52 ***	0.71 ***
Fisher correlation		*p* < 0.01		*p* < 0.01		*p* < 0.01	

- No significant correlation; *p*—significance: * *p* < 0.05 ** *p* < 0.01 *** *p* < 0.001.

**Table 3 children-10-01285-t003:** Comparison between Kindergarten and First Grade Regarding the Relationship Involving WM Components and Mathematics Skills.

	Counting		Relations		Arithmetic Skills		Number Sense	
	Kinder-Garten	Grade1	Kinder-Garten	Grade1	Kinder-Garten	Grade1	Kinder-Garten	Grade1
Simple Verbal Auditory WM	0.47 ***	0.69 **	0.57 ***	0.73 *	0.67 ***	0.67 ***	0.49 ***	-
Fisher correlation	*p* < 0.05		*p* < 0.05		*p* = 0.05		*p* < 0.01	
Simple Visual Spatial WM	0.46 ***	-	0.56 **	0.51 *	0.29 **	0.46 *	0.67 ***	0.83 *
Fisher correlation	*p* < 0.01		*p* < 0.05		*p* < 0.05		*p* < 0.01	
Complex WM	0.59 ***	0.86 **	0.54 **	0.89 *	0.61 ***	0.87 ***	0.63 ***	0.70 **
Fisher correlation	*p* < 0.01		*p* < 0.01		*p* < 0.01		*p* < 0.05	

- No significant correlation; *p*—significance: * *p* < 0.05 ** *p* < 0.01 *** *p* < 0.001.

## Data Availability

The data is part of a large longitudinal study; it will be available upon request.

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
