# Peer review of "The Role of Working Memory in Early Literacy and Numeracy Skills in Kindergarten and First Grade"

_children, 2023, doi:10.3390/children10081285_

Round 1

Reviewer 1 Report

Dear Authors,

Your manuscript contains very interesting and novel data; the study was well structured and scientifically grounded.   

Still, some questions and comments should be stated:

Lines 271-276:  Could you please explain why the age distribution in the 2nd assessment (7-10 years) was wider than in the 1st assessment (5-7 years)?

Line 329:  Some text is omitted after word e.g’.

Recommendation: I would suggest adding a sub-section “Procedure” in the section “Method”. In this sub-section, the design of assessment sessions could be presented more precisely; namely, the mode of tests presentation (presented individually or in a group? Assessed at school or in a lab?); the mean duration of a session; the number of sessions participants passed; the order of test presentation within a session (randomized or fixed order?) should be described.

Author Response

I would like to thank you for your review. The following correction were done according to your suggestions.
Line 271-276 - the age range was a mistake and corrected to the correct age of six to eight years old.
Line 329 - the e.g. was there by mistake and has been erased from the paper
As you suggested a section of procedure was add to the method section see lines 500-507.

Reviewer 2 Report

I begin by thanking you for the opportunity to review such an interesting work.

The manuscript is very pertinent, well-structured and with great potential to be published.

The abstract offers a very clear idea of the work.

The objectives are clearly defined and articulated with the methodological questions, which are very well defined and explained.

The results are clearly presented, analyzed, and discussed.

There is only one situation that should be reviewed and improved. The bibliography used is dated. Of the 130 references, only about 14% are from the last 5 years.

Thus, it is suggested that the theoretical framework be revised, including more updated bibliography and that this same recent bibliography be included in the discussion of the results.

Also, typos in the references should be corrected.

Author Response

I would like to thank you for your review. The following correction were done according to your suggestions.

I have updated the references with new papers I found, it is important to note that the theories on which the paper is based on (working memory, linguistic and numeric models) are the basic original model which were published more than five years ago but are still the main models in these fields and therefore the references are from older years). The new references have been added to the reference section. Furthermore some of the older references have been removed. I would like to emphasize that most references used are from the past 10 years. 

In addition the typos in the references have been corrected.

Round 2

Reviewer 2 Report

According to the guidelines of the journal, the cited references should be mainly from the last 5 years.

The review only added 3 articles from the last 5 years, and the problem initially identified remained. Understanding the position of the authors, mentioning that there are key theoretical references that should be mobilized, there are recent publications that help to understand the data obtained in a temporally better situated perspective.

Author Response

I have gone over all many new  references  and added another 10 new papers which are relevant to this study  from 2019 and newer and deleted some of the older references.  The older papers are mainly the basic theories and test references as I mentioned before.